# Eating Attitudes, Body Appreciation, Perfectionism, and the Risk of Exercise Addiction in Physically Active Adults: A Cluster Analysis

**DOI:** 10.3390/nu17132063

**Published:** 2025-06-20

**Authors:** Bettina F. Piko, Tamás L. Berki, Orsolya Kun, David Mellor

**Affiliations:** 1Department of Behavioral Sciences, University of Szeged, 6722 Szeged, Hungary; antonia.orsolya.kun@gmail.com; 2Department of Physical Education Theory and Methodology, Hungarian University of Sport Science, 1123 Budapest, Hungary; berki.tamas@tf.hu; 3School of Psychology, Deakin University, Burwood, VIC 3125, Australia; david.mellor@deakin.edu.au

**Keywords:** exercise addiction, eating attitudes, eating disorders, perfectionism, body image

## Abstract

**Background/Objectives**: Although regular physical exercise is protective for health, excessive engagement can contribute to the development of addiction. Further, the co-occurrence of exercise addiction (EA) and disordered eating (DE) is very frequent among athletes with several common risk factors. Our study focused on the associations between exercise addiction, eating attitudes, body appreciation, and perfectionism in a sample of physically active adults. **Methods**: Using a sample of Hungarian adults who were regular exercisers (*n* = 205, aged 18–70 years, mean age = 30.59 years; 77.1% females), cluster analysis was applied to identify participants’ profile according to their level of EA, DE attitudes, body appreciation, and dimensions of perfectionism. **Results**: Healthy exercisers had the second lowest level of EA and highest level of body appreciation, and they were not prone to DE (31.22%). Another cluster had a relatively low risk of EA but were potentially prone to DE, with poor body appreciation and a medium level of socially prescribed and other-oriented perfectionism (29.8%). Third, a group of exercisers was characterized by the highest risk of both EA and DE, who also reported relatively high levels of personal standards and organization (25.36%). Finally, those with the second highest risk of EA with a high tendency for dieting and bulimia and poor body appreciation were prone to socially prescribed and other-related perfectionism (13.66%). **Conclusions**: Symptoms of exercise addiction are not necessarily pathological, but they can serve as signals for the overuse of sports and undue achievement orientation, particularly when being associated with disordered eating attitudes.

## 1. Introduction

Regular physical exercise provides physiological and psychosocial protection against health problems [1,2,3]. Consistent with this, those who engage in regular exercise rate their own physical and mental health better than those who do not [4], and report a better quality of life [5]. Physical activity and healthy diet, together with positive mental health, provide three pillars of a well-balanced lifestyle to prevent chronic conditions such as obesity, diabetes, or cardiovascular disease [6,7].

Despite the benefits of physical exercise, excessive engagement can induce negative health outcomes such as an elevated risk of injury or chronic musculoskeletal disorders [8] and psychosocial problems (e.g., neglecting other parts of one’s life, such as work or school tasks) [9]. The challenge then is to establish the boundary between beneficial levels of exercise and health-impairing forms of exercise [10] because, in addition to the abovementioned potential negative outcomes, exercise can become addictive [11]. Although earlier studies have pointed out that exercise addiction may be a “good addiction” for athletes in that it helps them achieve their goals in competitions [12,13], it can be a “negative addiction” as well, e.g., among long distance runners who might lose control over their behavior [14,15]. Griffiths identified six components of exercise addiction [16]: salience (exercise dominates the individual’s life), mood modification (exercise results in positive change in one’s mood), tolerance (a need for an increasing amount of exercise to achieve the same psychological effect), conflict (interpersonal conflicts in one’s social network due to excessive exercise), withdrawal (unpleasant experiences caused by interrupting/reducing exercise), and relapse (return to previous amount of exercise after an injury). Based on these characteristics, its obsessive–compulsive nature can be recognized, similar to chemical addictions.

Further, De Coverley Veale suggested a distinction between primary and secondary exercise addiction (EA) [17], with the former being without a comorbid disorder (i.e., exercise is the primary goal) and the latter being an outcome of another dysfunction, most commonly disordered eating (DE), like anorexia nervosa, bulimia, or orthorexia nervosa (ON). While several studies included primary and secondary EA as independent conditions [18,19], other researchers have argued that EA cannot be viewed as irrespective of DE or body image dysfunctions. Instead, excessive exercise may serve as a tool for losing weight or achieving a more favorable body image [20]. Indeed, the co-occurrence of EA and DE is very frequent among athletes with several common risk factors, such as body dissatisfaction, disordered eating behaviors, or weight and shape concerns due to the internalization of body ideals or requirements of leanness in certain sports [21,22,23,24]. Body shape may mediate the relationship between DE and EA [25], supporting the concept of secondary EA. Other researchers have found that EA can be a primary addiction when the individual makes an effort to improve their sporting performance, elevate their well-being or self-esteem, or try to achieve a better body composition or weight loss without eating pathology [26]. However, the relationship between EA and DE is more complex in that individual and sociocultural pressures also influence both. An excessive amount of exercise or salience of sport may generate a need for perfectionism in terms of achievement, eating patterns, or appearance [27]. In a longitudinal study, ON (a drive to follow a healthy diet to the extreme) was predicted by perfectionism among exercisers [28]. Thus, searching for perfection may act as a common background variable of both EA and DE among athletes [29].

Perfectionism refers to cognitions characterized by not only setting high expectations or (often unrealistically) high standards for one’s achievement, but also the individual’s concerns with striving for perfection and flawlessness [30]. In the literature we also distinguish three types of trait perfectionism: self-oriented (setting high standards and self-critical appraisals for oneself); socially prescribed (a belief of others being critical in the case of one’s failure or not meeting the expected standards); and other-oriented (e.g., parents’ demand for perfection from their children or the coach’s demands from the athletes) [31]. Furthermore, researchers often differentiate between maladaptive perfectionism (concerns about mistakes and criticism) and adaptive perfectionism (striving to be organized to gain high achievement) [32]. The combination of these subtypes was found to have different roles in eating disorders, e.g., anorexia nervosa (AN) [33]. Meta-analyses provided evidence of the associations between both types of perfectionism and DE (e.g., binge eating, anorexia, and bulimia) [34,35]. Among the dimensions of trait perfectionism, socially prescribed and other-oriented perfectionism in particular have been found to predict ON [28], which has been particularly associated with exercise engagement, a common interest in a healthy lifestyle, and/or body weight control [36,37]. In another study, self-oriented perfectionism also had a direct positive effect on EA [38]. A recent study suggests that the type of passion in sport may have a different role in the relationship between perfectionism and DE: socially prescribed perfectionism is more likely to be associated with obsessive passion and disordered eating, while self-oriented perfectionism may be related to both obsessive and harmonious passion and well-being [39].

Edwards and Aron [40] suggest that there may be a complex interplay between perfectionism, exercise, disordered eating attitudes, and distorted body image or a lower level of body appreciation [41]. Therefore, in the current study, we investigated the associations between EA, DE attitudes (including dieting, bulimia, and oral control), ON (as a relatively newly described type of disordered eating), body appreciation (as a measure of body satisfaction), and dimensions of perfectionism. We applied cluster analysis for detecting groups of regular exercisers along their EA and ED scores in which we hypothesize that EA and DE may be differently associated depending on the dimensions of perfectionism (i.e., self-oriented, other-oriented, and socially prescribed) playing different roles in these clusters. We hope that the findings of this study will help inform not only athletes, but also members of the public about the risk of EA and DE, and provide practical support for recognizing their signs and associations—for example, their relationship with different forms of perfectionism.

## 2. Materials and Methods

### 2.1. Participants and Procedure

In this cross-sectional survey study, Hungarian citizens from Szeged and the metropolitan area were the participants. The study protocol was approved by the Institutional Review Board of the Doctoral School of Education, University of Szeged (No. 21/2023). Participants were recruited via a link posted on different social networking sites and sites for specific groups in which we expected high rates of participation from adults who were engaged in regular sports. The link led to an online questionnaire package hosted on Google Forms. The respondents were informed about the details of the study on the first page of the survey and could not progress to the main pages of the survey without providing informed consent. Participation was anonymous and voluntary. All participants were required to be at least 18 years of age and have an active lifestyle with regular physical activity (at least once a week, for a minimum of 30 min). Although there was no age limit, the only exclusion criterion was the lack of engagement in regular sports. Data were collected during a three-month period in the first quarter of 2024. A total of 205 participants (77.1% female, aged between 18 and 70 years old, mean age = 30.59 years; SD = 12.11 years) completed the questionnaire during the given time period. The higher participation of females is not unusual in voluntary surveys due to the greater willingness of females to participate [42].

### 2.2. Measures

#### 2.2.1. Exercise Addiction

The Hungarian validated version of the Exercise Addiction Inventory—Revised (EAI-R) [43,44] was used to measure levels of exercise addiction. This tool consists of six statements (e.g., “Exercise is the most important thing in my life”) reflecting Griffiths’ Components Model of Behavioral Addictions [16]. The respondents’ level of agreement was recorded on a 6-point Likert scale ranging from 1 (strongly disagree) to 6 (strongly agree). Responses to items were added to create a total scale score. Higher scores indicated a greater tendency for EA. Following the scale authors’ suggestion [45], scores in the top fifth (or above 80%) of the range of the scores, corresponding to a value above 24, were considered to reflect exercise addiction. Those with a score of 13 to 23 represent a potentially symptomatic group, while those with a score of 0 to 12 represent asymptomatic individuals. The Cronbach’s alpha coefficient of reliability in our sample was α = 0.79.

#### 2.2.2. Eating Attitudes

The validated Hungarian version [46] of the Eating Attitude Test-26 (EAT-26) was used to assess levels and risk of disordered eating habits [47]. The scale consists of 26 items (with one inverse item), and includes three subscales: EAT—Dieting (13 items, e.g., “Avoid foods with sugar in them”), EAT—Bulimia (6 items, e.g., “Vomit after I have eaten”), and EAT—Oral control (7 items, e.g., “Display self-control around food”). A six-point scale with response options ranging from “never” to “always” was used. Values of 3, 2, and 1 were given for “always”, “usually”, and “often”, respectively, and 0 for “sometimes”, “rarely”, and “never”). Higher scores reflect a greater tendency for showing these attitudes. Internal consistency values for the subscales were as follows: EAT—Dieting (α = 0.79), EAT—Bulimia (α = 0.70), EAT—Oral control (α = 0.60), and the reliability was 0.85 for the entire questionnaire.

#### 2.2.3. Orthorexic Behavior

We used the original Düsseldorf Orthorexie Scale [48,49] to capture orthorexic behavior. The participants responded to 10 items (e.g., “My thoughts constantly revolve around healthy nutrition and I organize my day around it”) and the response options ranged over a four-point scale: 1 (“this does not apply to me”) to 4 (“this applies to me”). Total scores may range from 10 to 40, with higher scores indicating greater orthorexic tendency. The Hungarian-translated version in this study demonstrated good internal consistency (Cronbach’s alpha = 0.80).

#### 2.2.4. Body Appreciation

Body appreciation was measured with the Hungarian validated version [50] of the Body Appreciation Scale-2 (BAS-2) [51], which comprises 10 items (e.g., “I respect my body”). Responses were recorded on a 5-point Likert-type scale of frequency ranging from 1 (never) to 5 (always). Total scores were used in the analyses, with higher scores reflecting higher levels of body appreciation. The Cronbach’s alpha coefficient of reliability in our sample was excellent (α = 0.96), similar to the level reported for the original version [51].

#### 2.2.5. Perfectionism

Perfectionism was measured with the Hungarian version of the Frost Multidimensional Perfectionism Scale (FMPS) [30]. The scale was adapted and previously applied on Hungarian populations [52]. It comprises 35 items designed to measure six dimensions of perfectionism: Concern over Mistakes (CM, 9 items, e.g., “People will probably think less of me if I make a mistake”), Doubts about Actions (DA, 4 items, e.g., “I usually have doubts about the simple everyday things I do”), Personal Standards (PS, 7 items, e.g., “I set higher goals than most people”), Parental Expectations (PE, 5 items, e.g., “My parents wanted me to be the best at everything”), Parental Criticism (PC, 4 items, e.g., “My parents never tried to understand my mistakes”) and Organization (O, 6 items, e.g., “I am a neat person”). Responses were recorded on a 5-point Likert-type scale measuring level of agreement ranging from 1 (strongly disagree) to 5 (strongly agree). Higher scores indicated higher tendencies to perfectionism in each dimension. We used four subscales suggested by Stöber (1998) [53]: Concern over mistakes and doubts about actions: CM + DA (Cronbach’s alpha = 0.92), Excessive concern with parents’ expectations and evaluation: PE + PC (Cronbach’s alpha = 0.92), Excessively high personal standards: PS (Cronbach’s alpha = 0.83), and Organization (Cronbach’s alpha = 0.89).

### 2.3. Data Analysis

Descriptive statistics and bivariate correlation were used to explore the study variables. K-means cluster analysis was used to identify participants’ profiles according to their level of exercise addiction, disordered eating attitudes, body appreciation, and dimensions of perfectionism. This cluster method has a distance-based clustering algorithm in which the distance is used as a measure of similarity. This method assigns cases to pre-specified subgroups by computing centroids and repeating steps to obtain the optimal centroid, that is, to achieve a stable cluster assignment [54]. The number of groups was defined based on the following analyses: elbow methods (for finding the optimal K value), Levene’s tests (for checking variance homogeneity), multivariate analysis of covariance (MANCOVA), and a set of ANOVA tests (with Tukey’s post hoc tests for justifying that all clusters were well-separated along these variables).

## 3. Results

### 3.1. Descriptive Statistics

Among the participants, 98 (47.8%) were white collar workers, 21 (10.2%) were blue collar workers, and 86 (42%) were students. Sample characteristics of their sporting activity are shown in Table 1. Most of the participants had been engaged in sporting activity for more than 10 years, and, on average, preferred to exercise 4 days or less a week (approximately 2 h per occasion).

Descriptive statistics for the study variables are presented in Table 2. Since some of the variables did not follow a normal distribution, K-means clustering was used in subsequent groupings of the sample.

### 3.2. Bivariate Associations

Table 3 presents the correlation matrix for the study variables. Among the correlates of exercise addiction, there was a positive relationship with EAT—Dieting [r (205) = 0.34, *p* < 0.001], EAT—Bulimia [r (205) = 0.27, *p* < 0.001], and EAT—Oral control [r (205) = 0.15, *p* < 0.05]. However, the strongest association was with ON [r (205) = 0.48, *p* < 0.001], which was also positively correlated with the other types of disordered eating (*p* < 0.001). The correlation with body appreciation was not significant (*p* > 0.05). Among the subscales of perfectionism, exercise addiction was positively associated with Concern over Mistakes and Doubts about Actions [r (205) = 0.21, *p* < 0.01], Personal Standards [r (205) = 0.25, *p* < 0.001], and Parental Expectations and Criticism [r (205) = 0.15, *p* < 0.05]. Personal Standards, Concern over Mistakes, and Doubts about Actions also showed positive correlations with all types of disordered eating. Parental Expectations and Criticism was slightly but positively correlated with dieting and bulimia [both r (205) = 0.17, *p* < 0.03]. It is also worth mentioning that body appreciation had a negative association with dieting [r (205) = −0.37, *p* < 0.001] and bulimia [r (205) = −0.29, *p* < 0.001]. However, the relationships between body appreciation and the dimensions of perfectionism varied: it was negatively correlated with Concern over Mistakes and Doubts about Actions [r (205) = −0.35, *p* < 0.001] as well as with the parental subscales (i.e., “maladaptive perfectionism) [r (205) = −0.22, *p* < 0.01], while it was positively correlated with Personal Standards [r (205) = 0.14, *p* < 0.05] and Organization (“adaptive perfectionism”) [r (205) = 0.27, *p* < 0.001].

### 3.3. Cluster Analysis

The participants’ profiles according to the level of exercise addiction, disordered eating attitudes, body appreciation, and dimensions of perfectionism were determined by conducting a K-means clustering analysis. The Elbow method identified the point (k = 4) for the optimal number of clusters, as adding more clusters only minimally decreased the total cluster variance. The means, standard deviations, z-scores, and F-values of the clusters are shown in Table 4, while cluster profiles based on z-scores are depicted in Figure 1. Although none of the clusters’ mean scores for EA reached the cut-off value (above 24.0), cluster 4 was closest (approx. 23). Thus, we can only discuss different levels of potential risk.

Cluster 1 included 61 participants (29.76%) with the lowest level of the exercise addiction scale. Compared with the other clusters, they showed a medium level of disordered eating patterns including orthorexic tendencies, and a relatively low degree of body appreciation. They reported the lowest level of organization and personal standards but a medium level of socially prescribed and other-oriented perfectionism. Therefore, we can describe this cluster as having a relatively low risk of EA but potentially being prone to DE, with an inappropriate body appreciation and a sort of socially prescribed and other-oriented perfectionism.

Cluster 2 included 28 participants (13.66%) with the second highest level of exercise addiction and a high tendency for dieting and bulimia. Their body appreciation was also relatively poor. Among the dimensions of perfectionism, these individuals reported the highest levels of concern over mistakes and doubts about actions, as well parental expectations and criticism (socially prescribed and other-related perfectionism). Thus, we can describe this cluster as being characterized by socially prescribed and other-oriented perfectionists with a potential risk of EA, and a disposition to dieting due to their low level of body appreciation.

Cluster 3 consisted of 64 participants (31.22%) characterized by the highest level of body appreciation and the second lowest level of EA. These individuals were neither prone to disordered eating nor perfectionism, except for a relatively higher level of organization. In terms of their eating patterns, dieting has the lowest rate among them. Therefore, these participants seem to be “healthy exercisers” who are well organized and satisfied with their body image.

Finally, Cluster 4 included 52 participants (25.36%) with the highest levels of EA and DE (namely, dieting, oral control, ON) except for bulimia. Although they showed a relatively high level of body appreciation, these individuals possessed high levels of self-oriented perfectionism (personal standards and organization). This cluster represents the highest risk of EA with the greatest proneness to DE in which the self-oriented dimension of perfectionism might play a decisive role.

The 22 participants who chose “competitive” as their level of sport were grouped in each cluster in the following order: most of them belonged to Cluster 4 (*n* = 12), five belonged to Cluster 3, three of them were part of Cluster 2, and two were categorized as being part of Cluster 1.

### 3.4. Multivariate Analysis

In our second step, multivariate analyses were conducted to justify our results. A MANOVA showed a significant result (Wilk’s Λ = 0.85, F10) = 15.55, *p* < 0.001, η^2^p = 0.32). Due to the significant result, an ANOVA was used in further analyses. Table 4 includes the means, standard deviations, z-scores, and F-values for the four clusters. All variables showed significant differences and the partial eta squared varied between 0.04 and 0.67.

The Tukey post hoc tests indicated that in the case of EA, Cluster 2 > Cluster 1 (*p* = 0.005), Cluster 3 > Cluster 1 (*p* = 0.042), Cluster 4 > Cluster 1 (*p* < 0.001), and Cluster 4 > Cluster 3 (*p* = 0.001). In terms of EAT–Dieting, Cluster 2 > Cluster 1 (*p* = 0.040), Cluster 2 > Cluster 3 (*p* = 0.001), Cluster 4 > Cluster 3 (*p* < 0.001), Cluster 4 > Cluster 1 (*p* = 0.002). According to EAT–Bulimia, Cluster 2 > Cluster 1 (*p* = 0.019), Cluster 2 > Cluster 3 (*p* = 0.001), and Cluster 4 > Cluster 3 (*p* < 0.001). EAT–Oral Control showed marginally different values only between Cluster 1 and Cluster 4 (*p* = 0.059). In terms of Orthorexia, Cluster 4 differed significantly from all other clusters: Cluster 4 > Cluster 1 (*p* < 0.001), Cluster 4 > Cluster 2 (*p* = 0.002), and Cluster 4 > Cluster 3 (*p* < 0.001). Cluster 1 showed significantly lower levels of the body appreciation scale from Cluster 3 and Cluster 4 (*p* < 0.001); likewise, Cluster 2 was also lower in levels than Cluster 3 and Cluster 4 (*p* < 0.001). In addition, Cluster 3 > Cluster 4 (*p* = 0.014). For Concern over Mistakes and Doubts about Actions (CM + DA) and Excessive Concern with Parents’ Expectations and Evaluation (PE + PC), all clusters significantly differed from the others (*p* = 0.001 or below). In terms of Personal Standards, Cluster 2 > Cluster 1 and Cluster 4 > Cluster 1 (*p* < 0.001); Cluster 4 > Cluster 3 (*p* < 0.001) and Cluster 2 > Cluster 3 (*p* < 0.001). Finally, in terms of Organization, Cluster 4 > Cluster 1 (*p* < 0.001), Cluster 3 > Cluster 1 (*p* = 0.002); however, the difference between Cluster 1 and Cluster 2 was not significant (*p* = 0.061).

## 4. Discussion

The aim of this study was to detect regular exercisers’ profiles based on the associations between exercise addiction (EA), disordered eating attitudes (DE), body image, and perfectionism, using cluster analysis. This analysis allowed us to test a complex interplay between these variables and to explore the roles of their different dimensions or subtypes [40]. To the best of our knowledge, this is the first study examining EA, ED, and other relevant psychological factors among a sample of adults who engage in regular physical activity. Using K-means clustering analysis, four user profiles emerged in our study, which identified a number of specific associations between EA and its correlates. Previous studies explored a frequent concomitance between EA and DE [23,24]. Besides their mutual interaction, this co-occurrence may stem from common antecedents, such as low levels of body appreciation [41] or dissatisfaction with body shape [25], or different dimensions of perfectionism [28,29,39]. We aim to explore these associations in a sample of physically active adults and not only among competitive athletes.

The largest portion of the sample (31.22%) belonged to Cluster 3, which was considered to be “healthy exercisers” who showed the greatest appreciation of their body and scored relatively higher on the Organization dimension of perfectionism. They showed the second lowest level of EA and they were not prone to DE, perhaps due to body satisfaction and the adaptive (self-oriented) dimension of perfectionism. The potentially protective roles of these psychological factors are in concordance with findings of previous research [39,41]. Adaptive perfectionism involves striving for high but realistic goals rather than setting unrealistic standards and being afraid of making mistakes [31,33]. It seems that adaptive perfectionism does not impede the appreciation of one’s body.

The second largest group, Cluster 1 (29.76%), included participants with the lowest levels of EA but who were moderately prone to DE (specifically to dieting and bulimia but not to ON and oral control). They also scored low on the body appreciation scale and two dimensions of perfectionism: organization and personal standards (i.e., self-oriented perfectionism) with a medium level of socially described and other-oriented perfectionism. These individuals were not satisfied with their body and had a drive to diet (sometimes up to bulimia) to achieve a “more perfect” body. However, this drive did not stem from self-oriented (more adaptive) perfectionism (i.e., striving for being organized), but more from socially prescribed (less adaptive) perfectionism (i.e., concern about others’ evaluation). This finding further supports the different roles of self-oriented perfectionism and other-oriented/socially prescribed perfectionism [39]. A previous study found that socially prescribed and other-oriented perfectionism were more relevant in the development of DE [28].

The next cluster (25.36%), Cluster 4, consisted of people with the highest levels of EA and DE (namely, dieting, oral control, and ON) as well as relatively high levels of personal standards and organization. Interestingly, their level of body appreciation was the second highest, indicating that they see their own body in a realistic way, despite any dieting activity. The co-occurrence of EA and DE is previously well established [23,24]. Our results suggest that when people are susceptible to EA and DE, self-oriented perfectionism (a drive for being organized to achieve well) is not adaptive at all. In addition, levels of socially prescribed and other-oriented perfectionism also showed the second greatest values compared with other clusters. This finding is in line with previous research results [28,39]. Since a high level of DE is often linked with excessive exercise among competitive athletes, the internalization of body ideals (“leanness”) may be socially prescribed for them [13,21,22,27]. Striving for high (“perfect”) achievement can be either self-oriented or other-oriented; however, self-oriented perfectionism seems to have a direct positive effect on EA in order to strengthen one’s self-esteem [38].

Finally, the smallest group (13.66%), Cluster 2, is characterized by the second highest score on EA associated with a high tendency toward bulimia (and dieting). Besides their poor body appreciation, these participants reported the highest levels of the socially prescribed and other-related (maladaptive) dimensions of perfectionism. We propose that this type of perfectionism may play a role in their ideas about a perfect body, and also that they may use exercise as a tool to achieve this ideal (secondary EA) [17,19].

These findings support the concept that there are important associations among these variables, depending on the levels of EA, the type of DE, and the dimensions of perfectionism. When there is ample evidence of the risk for EA, exercisers are prone to dieting, oral control, and ON, while in the case of a moderate risk of EA, the development of bulimia has a higher probability (which may lead to secondary EA). However, in the previous group, self-oriented perfectionism had higher scores; in the latter group, socially prescribed and other-oriented types of perfectionism were decisive with a low level of body appreciation. Furthermore, when a similar moderate level of EA was related to a greater level of body appreciation and self-oriented perfectionism, this type of perfectionism (namely, a drive to be well organized) seemed adaptive to avoid DE. In other words, a complex interplay among the variables may lead to different outcomes in which sometimes self-oriented perfectionism may even serve as a protection against EA, but only when people appreciate their body.

These findings may add relevant information to the literature on EA and DE. While most studies on this issue only include competitive athletes, we know less about these tendencies in general populations. However, due to the cross-sectional study design, cause-and-effect relationships cannot be implied. Furthermore, level of exercising was only based on self-reporting, and groups did not demonstrate high levels of addiction. This is not surprising, given the nature of our sample: we focused on the physically active general population instead of athletes at a higher level of sports. Further, the great variability of sporting activity did not allow us to involve types of sporting in the analyses, or make the difference between competitive vs. recreational sport type. In addition, the relatively small sample size and the female-biased sample may influence the generalizability of the findings. While Frost’s concept of perfectionism has been widely used (particularly in Hungary, being the only scale translated), a more recent model of the dynamic interplay of perfectionist striving and perfectionism concerning [55] may be also relevant to map a different viewpoint to these associations. Finally, the reliability of EAT oral control subscale proved relatively low (<0.70), similar to other Hungarian studies [56]. It seems that the factorial structure of this subscale appears to be less robust compared with the other two subscales [57]. Further adaptation may be necessary to improve it in different cultural contexts.

Future studies may involve a greater (and more gender-balanced) sample size and apply latent profile analysis to detect the role of combined dimensions of perfectionism. Other variables may also be useful to include, for example, other personality and mental health measures (e.g., anxiety, neuroticism, more types of addictions) or a detailed description of sports types. In addition, identifying protective factors to avoid EA and DE would be helpful for preventive purposes, such as resilience factors or coping with stress.

## 5. Conclusions

Our findings draw attention to EA tendencies that can be present not only among competitive athletes, but also in physically active adults who engage in regular sports. Likewise, DE can be linked to EA in this population when certain psychological symptoms occur. Body image and perfectionism seem to have important roles in the interplay between EA and DE, and we need to further clarify their complex associations. EA is often viewed as a socially accepted behavior; however, when it is associated with DE, concerns about body image, and perfectionism, we should be mindful of potential impacts on well-being and mental health. Symptoms of EA are not necessarily pathological, but they can act as signals for overuse of exercise and undue achievement orientation, particularly when being associated with disordered eating attitudes. We hope that professionals, including leisure activity leaders, personal trainers, coaches, and others, can incorporate this knowledge into their practice to help support their clients. For example, by understanding the signs and symptoms of EA, they may incorporate client-centered approaches and promote balanced fitness habits. This includes using screening tools, offering or helping with seeking support, and fostering a positive relationship with exercise that prioritizes well-being over excessive training. Those who engage in regular sporting activity may also seek help for a healthy diet from dieticians or nutrition/lifestyle specialists and avoid drastic dietary habits.

## Figures and Tables

**Figure 1 nutrients-17-02063-f001:**
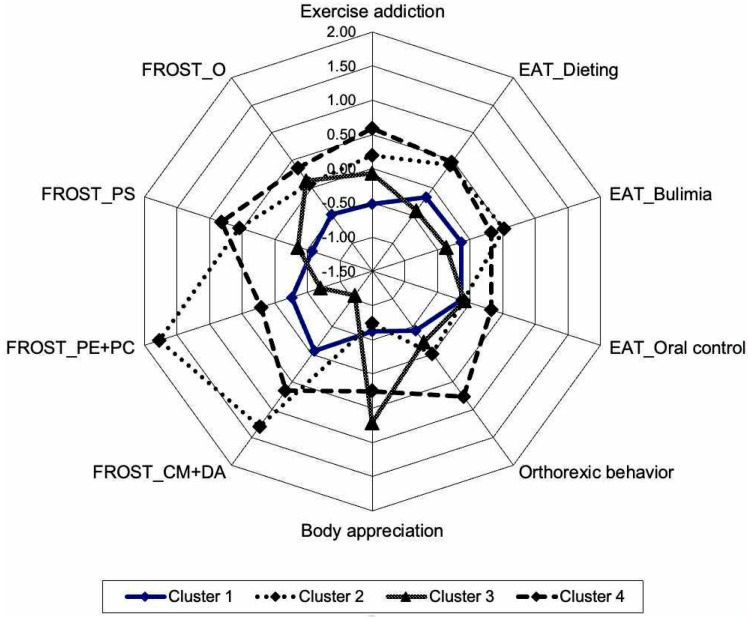
Cluster profiles based on z-scores. EAT = Eating Attitudes Test, O = Organization, PS = Personal Standards, PE = Parental Expectations, PC = Parental Criticism, CM = Concern over Mistakes, DA = Doubts about Actions, FROST = Frost perfectionism scale.

**Table 1 nutrients-17-02063-t001:** Descriptive statistics for sample characteristics of sociodemographics and sporting activity.

Variables	*n* (%)
Sex	
Male	47 (22.9)
Female	158 (77.1)
Employment status	
White collar	98 (47.8)
Blue collar	21 (10.2)
Student	86 (42.0)
Length of exercise time (years)	
1 year or less	40 (19.5)
2–3 years	24 (11.7)
3–5 years	17 (8.3)
5–10 years	37 (18.0)
>10 years	87 (42.4)
Exercise on days/week	
1–2 days	86 (42.0)
3–4 days	90 (43.9)
5 days of more	29 (14.1)
Exercise time per occasion	
1 h or less	72 (35.1)
1–2 h	114 (55.6)
3–5 h	16 (7.8)
>5 h	3 (1.5)
More than one occasion a day	
No	126 (61.5)
Yes	79 (38.5)
Level of exercise	
Hobby	183 (89.3)
Competitive	22 (10.7)

**Table 2 nutrients-17-02063-t002:** Descriptive statistics for the study scales.

	Min–Max	Mean (SD)	Skewness	Kurtosis
1. Exercise addiction	6–35	19.30 (6.16)	0.13	−0.35
2. EAT—Dieting	0–31	6.27 (6.38)	1.35	1.50
3. EAT—Bulimia	0–11	0.96 (1.86)	2.69	8.20
4. EAT—Oral control	0–18	2.31 (2.51)	2.65	11.31
5. Orthorexic behavior	10–33	18.29 (5.04)	0.60	0.18
6. Body appreciation	15–50	35.60 (8.05)	−0.54	−0.26
7. Concern over Mistakes + Doubts about Actions	13–65	36.53 (12.31)	0.10	−0.62
8. Parental Expectations + Parental Criticism	9–45	20.28 (9.14)	0.74	−0.21
9. Personal Standards	9–35	23.90 (6.12)	−0.18	−0.75
10. Organization	6–30	23.87 (4.81)	−0.78	0.33

**Table 3 nutrients-17-02063-t003:** Correlation matrix for the study scales.

	1	2	3	4	5	6	7	8	9	10
1. Exercise addiction	(0.79) #									
2. EAT—Dieting	0.34 ***	(0.79)								
3. EAT—Bulimia	0.27 ***	0.64 ***	(0.70)							
4. EAT—Oral Control	0.15 *	0.20 **	0.31 ***	(0.60)						
5. Orthorexic behavior	0.48 ***	0.45 ***	0.29 ***	0.19 **	(0.80)					
6. Body appreciation	0.05	−0.37 ***	−0.29 ***	0.09	0.07	(0.96)				
7. Concern over Mistakes + Doubts about Actions	0.21 **	0.32 ***	0.35 ***	0.13	0.27 ***	−0.35 ***	(0.92)			
8. Parental Expectations + Parental Criticism	0.15 *	0.17 *	0.17 *	0.01	0.11	−0.22 **	0.58 ***	(0.92)		
9. Personal Standards	0.25 ***	0.24 **	0.23 **	0.24 **	0.39 ***	0.14 *	0.51 ***	0.31 ***	(0.83)	
10. Organization	0.12	0.09	0.07	0.21 **	0.20 **	0.27 ***	0.09	0.01	0.53 ***	(0.89)

Notes. Correlation coefficients, * *p* < 0.05 ** *p* < 0.01 *** *p* < 0.001, # Cronbach’s alpha.

**Table 4 nutrients-17-02063-t004:** Means, SD, z-scores, and F-test for the scales of exercise addiction, eating attitudes, body appreciation, and dimensions of perfectionism in the final clusters.

	CLUSTER 1 Mean (SD) z-Score	CLUSTER 2 Mean (SD) z-Score	CLUSTER 3 Mean (SD) z-Score	CLUSTER 4 Mean (SD) z-Score	F-Value	η^2^p
Exercise addiction	16.11 (4.61) −0.52	20.46 (6.61) 0.19	18.86 (6.17) −0.07	22.96 (5.46) 0.59	14.36 **	0.18
EAT—Dieting	5.20 (5.58) −0.17	8.93 (7.42) 0.42	3.67 (3.64) −0.41	9.29 (7.59) 0.47	11.03 **	0.14
EAT—Bulimia	0.72 (1.51) −0.13	1.93 (2.24) 0.52	0.28 (0.72) −0.36	1.54 (2.52) 0.32	8.15 **	0.11
EAT—Oral Control	1.95 (1.89) −0.14	2.07 (2.87) −0.10	2.09 (1.81) −0.09	3.13 (3.42) 0.33	2.58 *	0.04
Orthorexic behavior	16.20 (3.95) −0.42	18.29 (5.02) −0.001	17.16 (4.11) −0.22	22.15 (5.21) 0.77	18.57 **	0.22
Body appreciation	30.59 (6.93) −0.62	29.61 (7.76) −0.74	41.36 (5.51) 0.71	37.63 (6.17) 0.25	39.07 **	0.37
Concern over Mistakes + Doubts about Actions	35.84 (7.01) −0.06	52.71 (8.31) 1.31	23.52 (6.01) −1.06	44.63 (6.55) 0.66	156.12 **	0.67
Parental Expectations + Parental Criticism	17.90 (6.29) −0.26	36.54 (5.06) 1.78	13.89 (5.16) −0.70	22.17 (5.88) 0.21	108.57 **	0.62
Personal Standards	20.29 (4.89) −0.57	27.18 (4.51) 0.54	21.72 (5.72) −0.36	28.92 (4.26) 0.82	35.97 **	0.35
Organization	21.54 (5.24) −0.48	24.25 (4.73) 0.08	24.53 (4.53) 0.14	25.60 (3.60) 0.36	8.25 **	0.11
Percentage (*n*)	29.76% (61)	13.66% (28)	31.22% (64)	25.36% (52)		

Notes. z-score = standardized score, η^2^p = partial eta squared (effect size), F-value derived from F-test in ANOVA, * *p* < 0.05, ** *p* < 0.001.

## Data Availability

The dataset generated and analyzed during the current study is available at https://osf.io/j53vf/?view_only=a567c06891bd4d44a1ef7b5bf2d4b0a8 (accessed on 22 November 2024).

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
