# Peer review of "Eating Attitudes, Body Appreciation, Perfectionism, and the Risk of Exercise Addiction in Physically Active Adults: A Cluster Analysis"

_nutrients, 2025, doi:10.3390/nu17132063_

Round 1
Reviewer 1 Report
Comments and Suggestions for Authors
This is an important area of research informing the care and support physically active people at risk of exercise addiction. Thank you for the chance to read this very interesting manuscript. Comments below are offered for your consideration to enhance the manuscript and points 2 and 12 are aimed at increasing readership and citation of the final paper.
1. Abstract: Please rephrase this part of the sentence using a precise scientific term rather than "a sort of": "...a sort of socially prescribed and other-oriented perfectionism (29.8%).
2. Introduction: For the health professionals who will read the paper, please add a sentence to the last paragraph of the introduction to explain how the new knowledge from this study might inform practical support for athletes and members of the public at risk of exercise addition or eating disorder.
3. Under 2.2.5 Perfectionism. The first sentence appears to to have the words Body appreciation accidentally inserted.
4. Table 3. The legend under table 3: "Notes. Correlation coefficients. *p < 0.05 **p < 0.01 #Cronbach’s alpha." appears to be missing the explanation of the symbol ***p < 0.001
5. Figure 1. Please state what FROST stands for in the legend. It might also be worth adding it below FMPS in the list of abbreviations.
6. Page 8, line 260, As in the abstract, please find a better way of expressing "a sort of".
6. Page 9, lines 277-8, Please review this phrase of the sentence which does not sound natural in English: "Although they relatively well appreciated their body,..."
7. It would be very interesting to know how many of the 22 people who chose Competitive for level of sport were grouped in each cluster, especially cluster 4. Please could this be added to the section on cluster 4 on page 9?
8. Please add the cluster numbers on page 10 in the text summarising the profiles and associations. For example line 320, ""The largest portion of the sample (31.22%) belonged to the cluster considered.." might be amended to "The largest portion of the sample (31.22%) belonged to cluster 3, which was considered to be..."
9. Page 10, lines 359-60, "We assume that this type of perfectionism may play a role in their ideas about a perfect body" Please consider replacing the word assume, for example with hypothesise /or propose /or suggest.
10. Page 11, Line 371, "In a word, a.." replace with "In other words..."
11. Page 11, lines 382-3 "...or make difference between..." Please insert the between make and difference: "...or make the difference between..."
12. Page 11, While it is correctly acknowledged that there is more work to be done, please add a little text to the discussion to expand upon this point in the conclusion: "Symptoms of EA are not necessarily pathological but they can act as signals for overuse of sports and undue achievement orientation, particularly when being associated with disordered eating attitudes." to indicate to readers how professionals, including leisure activity leaders, personal trainers, coaches and others might incorporate this knowledge into their practice to help support their clients:
Author Response
Manuscript ID: nutrients-3699766
Reviewer 1
Dear Reviewer,
Thank you for your remarks; here are our replies to each of them:
This is an important area of research informing the care and support physically active people at risk of exercise addiction. Thank you for the chance to read this very interesting manuscript. Comments below are offered for your consideration to enhance the manuscript and points 2 and 12 are aimed at increasing readership and citation of the final paper.
- Abstract: Please rephrase this part of the sentence using a precise scientific term rather than "a sort of": "...a sort of socially prescribed and other-oriented perfectionism (29.8%).
RE: Rephrased.
- Introduction: For the health professionals who will read the paper, please add a sentence to the last paragraph of the introduction to explain how the new knowledge from this study might inform practical support for athletes and members of the public at risk of exercise addition or eating disorder.
RE: Thank you for this suggestion. We have added a sentence to the last paragraph.
- Under 2.2.5 Perfectionism. The first sentence appears to to have the words Body appreciation accidentally inserted.
RE: Thank you for this note. We have deleted them.
- Table 3. The legend under table 3: "Notes. Correlation coefficients. *p < 0.05 **p < 0.01 #Cronbach’s alpha." appears to be missing the explanation of the symbol ***p < 0.001
RE: Thank you for this note. We have added the explanation of this symbol.
- Figure 1. Please state what FROST stands for in the legend. It might also be worth adding it below FMPS in the list of abbreviations.
RE: We have added them as requested.
- Page 8, line 260, As in the abstract, please find a better way of expressing "a sort of".
RE: Rephrased.
- Page 9, lines 277-8, Please review this phrase of the sentence which does not sound natural in English: "Although they relatively well appreciated their body,..."
RE: Thank you for this note. We have rephrased this sentence.
- It would be very interesting to know how many of the 22 people who chose Competitive for level of sport were grouped in each cluster, especially cluster 4. Please could this be added to the section on cluster 4 on page 9?
RE: Thank you for this suggestion: we really think it is a good idea. We have added this information to the last paragraph.
- Please add the cluster numbers on page 10 in the text summarising the profiles and associations. For example line 320, ""The largest portion of the sample (31.22%) belonged to the cluster considered.." might be amended to "The largest portion of the sample (31.22%) belonged to cluster 3, which was considered to be..."
RE: We have added cluster numbers.
- Page 10, lines 359-60, "We assume that this type of perfectionism may play a role in their ideas about a perfect body" Please consider replacing the word assume, for example with hypothesise /or propose /or suggest.
RE: We have changed the word "assume" with "propose" as requested.
- Page 11, Line 371, "In a word, a.." replace with "In other words..."
RE: Replaced.
- Page 11, lines 382-3 "...or make difference between..." Please insert the between make and difference: "...or make the difference between..."
RE: Inserted.
- Page 11, While it is correctly acknowledged that there is more work to be done, please add a little text to the discussion to expand upon this point in the conclusion: "Symptoms of EA are not necessarily pathological but they can act as signals for overuse of sports and undue achievement orientation, particularly when being associated with disordered eating attitudes." to indicate to readers how professionals, including leisure activity leaders, personal trainers, coaches and others might incorporate this knowledge into their practice to help support their clients.
RE: Thank you for this suggestion, this is a very good idea. We have completed the last paragraph with some concrete suggestions.
Thank you very much for your help in improving the quality of our paper.
We really hope that based on your suggestions our paper has much improved and now it is suitable for publishing.
15.06.2025
Authors
Reviewer 2 Report
Comments and Suggestions for Authors
Interesting idea of ​​this study, my recommendations are the following:
Abstract - I recommend that instead of body appreciation, body image be used.
2.1. I recommend mentioning the exclusion criteria from the sample.
2.2.2. Eating attitudes – I recommend mentioning the reliability for the entire questionnaire, which is much more specific. Also for EAT−Oral control (alfa Cronbach = 0.60), the reliability is very low, most researchers considering that the minimum limit value is 0.70. I recommend clarifications as to why this resulting value is so low.
Table 2 when numbering the indicators in the first column, there is a mistake, skip from 9 to 12, I recommend correction.
Table 4 I recommend that under the table, a descriptive description of what the acronyms and symbols represent is mentioned.
Lines 318-319 mention that the sample does not include competitive athletes, but according to table 1 you mention the Level of sporting indicator: hobby and competitive. Also 22 of the subjects are included in competitive, I recommend clarification.
I recommend expanding the aspects regarding future research directions, at the end of the Discussion section.
I also recommend mentioning the practical implications of the study.
Author Response
Manuscript ID: nutrients-3699766
Reviewer 2
Dear Reviewer,
Thank you for your remarks; here are our replies to each of them:
Interesting idea of ​​this study, my recommendations are the following:
Abstract - I recommend that instead of body appreciation, body image be used.
RE: Thank you for this note – I am sorry to say but we have applied the term (also the concept and the name of the scale) "body appreciation", at this stage of the study we cannot change it with another term.
2.1. I recommend mentioning the exclusion criteria from the sample.
RE: We have added information on exclusion criteria.
2.2.2. Eating attitudes – I recommend mentioning the reliability for the entire questionnaire, which is much more specific. Also for EAT−Oral control (alfa Cronbach = 0.60), the reliability is very low, most researchers considering that the minimum limit value is 0.70. I recommend clarifications as to why this resulting value is so low.
RE: We agree with the Reviewer. We added the reliability for the entire questionnaire as well. Also, we clarified the low reliability for oral control (among the limitations).
Table 2 when numbering the indicators in the first column, there is a mistake, skip from 9 to 12, I recommend correction.
RE: Thank you for this note. We have corrected it.
Table 4 I recommend that under the table, a descriptive description of what the acronyms and symbols represent is mentioned.
RE: We have added a description of the symbols.
Lines 318-319 mention that the sample does not include competitive athletes, but according to table 1 you mention the Level of sporting indicator: hobby and competitive. Also 22 of the subjects are included in competitive, I recommend clarification.
RE: Thank you for this note – we agree with it. We have clarified this issue.
I recommend expanding the aspects regarding future research directions, at the end of the
Discussion section.
RE: Thank you for this note, we have added some other suggestions for future research.
I also recommend mentioning the practical implications of the study.
RE: At the end of the last paragraph, we added some practical implications.
Thank you very much for your help in improving the quality of our paper.
We really hope that based on your suggestions our paper has much improved and now it is suitable for publishing.
15.06.2025
Authors